# Reduction in Cold Stress in an Innovative Metabolic Cage Housing System Increases Animal Welfare in Laboratory Mice

**DOI:** 10.3390/ani13182866

**Published:** 2023-09-09

**Authors:** Laura Wittek, Chadi Touma, Tina Nitezki, Thomas Laeger, Stephanie Krämer, Jens Raila

**Affiliations:** 1Department of Physiology and Pathophysiology of Nutrition, Institute of Nutritional Science, University of Potsdam, 14558 Nuthetal, Germanythomas.laeger.1@uni-potsdam.de (T.L.); jens.raila@uni-potsdam.de (J.R.); 2Department of Behavioural Biology, Osnabruck University, 49076 Osnabruck, Germany; chadi.touma@uni-osnabrueck.de; 3Interdisciplinary Center of 3Rs in Animal Research (ICAR3R), Clinic of Veterinary Medicine, Justus Liebig University of Giessen, 35392 Giessen, Germany; stephanie.kraemer@vetmed.uni-giessen.de

**Keywords:** metabolic cage, laboratory mice, refinement, animal welfare

## Abstract

**Simple Summary:**

Mice can be housed in metabolic cages in order to answer specific scientific questions. The base of this cage system consists of a metal wire mesh in order to collect urine and feces of mice held therein. The food and water intake can also be assessed. The housing of mice in metabolic cages is considered highly stressful and cold stress is also often addressed because these social animals are single-housed and no further materials (e.g., nesting material) are provided to them. Therefore, in this study, we aimed to reduce stress during metabolic cage housing for mice. An innovative metabolic cage was constructed with integrated cage design improvements and was directly compared with a commercially available metabolic cage and a control cage. The following parameters were selected to evaluate the stress of laboratory mice: body weight, body composition, food intake, cage and body surface temperature, messenger ribonucleic acid (mRNA) expression of uncoupling protein 1 (*Ucp1*) in brown adipose tissue (BAT), fur score, and fecal corticosterone metabolites (CMs). Our results clearly show that metabolic cage housing elicits a stress response in mice, which is related to the cold housing conditions. The innovative metabolic cage represents a first attempt to reduce cold stress during metabolic cage housing.

**Abstract:**

Housing in metabolic cages can induce a pronounced stress response. Metabolic cage systems imply housing mice on metal wire mesh for the collection of urine and feces in addition to monitoring food and water intake. Moreover, mice are single-housed, and no nesting, bedding, or enrichment material is provided, which is often argued to have a not negligible impact on animal welfare due to cold stress. We therefore attempted to reduce stress during metabolic cage housing for mice by comparing an innovative metabolic cage (IMC) with a commercially available metabolic cage from Tecniplast GmbH (TMC) and a control cage. Substantial refinement measures were incorporated into the IMC cage design. In the frame of a multifactorial approach for severity assessment, parameters such as body weight, body composition, food intake, cage and body surface temperature (thermal imaging), mRNA expression of uncoupling protein 1 (*Ucp1)* in brown adipose tissue (BAT), fur score, and fecal corticosterone metabolites (CMs) were included. Female and male C57BL/6J mice were single-housed for 24 h in either conventional Macrolon cages (control), IMC, or TMC for two sessions. Body weight decreased less in the IMC (females—1st restraint: −6.94%; 2nd restraint: −6.89%; males—1st restraint: −8.08%; 2nd restraint: −5.82%) compared to the TMC (females—1st restraint: −13.2%; 2nd restraint: −15.0%; males—1st restraint: −13.1%; 2nd restraint: −14.9%) and the IMC possessed a higher cage temperature (females—1st restraint: 23.7 °C; 2nd restraint: 23.5 °C; males—1st restraint: 23.3 °C; 2nd restraint: 23.5 °C) compared with the TMC (females—1st restraint: 22.4 °C; 2nd restraint: 22.5 °C; males—1st restraint: 22.6 °C; 2nd restraint: 22.4 °C). The concentration of fecal corticosterone metabolites in the TMC (females—1st restraint: 1376 ng/g dry weight (DW); 2nd restraint: 2098 ng/g DW; males—1st restraint: 1030 ng/g DW; 2nd restraint: 1163 ng/g DW) was higher compared to control cage housing (females—1st restraint: 640 ng/g DW; 2nd restraint: 941 ng/g DW; males—1st restraint: 504 ng/g DW; 2nd restraint: 537 ng/g DW). Our results show the stress potential induced by metabolic cage restraint that is markedly influenced by the lower housing temperature. The IMC represents a first attempt to target cold stress reduction during metabolic cage application thereby producing more animal welfare friendlydata.

## 1. Introduction

Metabolic cages represent a useful tool to collect metabolic data of individual laboratory mice including both the collection and separation of urinary and fecal samples and the assessment of food and water intake. Metabolic cage housing is broadly applied in the field of pharmacokinetic and pharmacodynamic studies, experimental nephrology, and studies of gastrointestinal function, among others [1,2]. Since mice are single-housed, enrichment, bedding, and nesting materials are lacking, and the bottom of the cage is made out of a metal wire mesh, metabolic cages induce a pronounced stress response during restraint [1,2]. Whether mice are able to adjust to metabolic cage housing is questionable, because studies have proven that mice, in contrast to rats, are sensitive to low-level stressors [1,3]. One study demonstrated that male BALB/c mice do not acclimatize to a metabolic cage after 3 weeks while plasma and urinary parameters stabilized in female and male C57BL/6J mice after a 3 d restraint in a metabolic cage within the scope of another study [1,2]. Mice have a thermoneutral zone at approximately 30 °C [4,5,6]. The thermoneutral zone defines the temperature range at which heat production from basal energy expenditure is sufficient to compensate for heat losses resulting from the existent temperature gradient between room and body temperature [4,5].

Cold stress therefore describes temperatures below the thermoneutral zone resulting in an increase in energy demand for body temperature maintenance [7]. How laboratory mice cope with standardized room temperatures at 20–22 °C in light of potential cold stress is often addressed. Interestingly, when comparing the resting metabolic rate with energy expenditure, mice kept in thermoneutral conditions (30 °C) show a 1.6 times higher energy expenditure compared to the resting metabolic rate, whereas for mice kept in standard conditions (21 °C) the ratio between energy expenditure and resting metabolic rate increases to 2.6 and 3.5 for day- and nighttime, respectively [8].

The energy expenditure of mice may be elevated particularly during metabolic cage housing at 20–22 °C due to the absence of conspecifics and nesting material as mice are deprived of the possibility of keeping themselves warm through huddling and nests [9,10]. The heat of mice is additionally dissipated from the paws via the metal grid [11,12]. To investigate potential cold stress during metabolic cage restraint, thermal images of mice housed in these cage systems may be taken. Thermal imaging represents an alternative to e.g., rectal thermometry since handling and fixation of mice is not required, omitting potential stress-induced hyperthermia [13,14]. Infrared thermography represents an efficacious tool to “quantify the skin’s surface temperature based on visualizations of thermographic changes” [15]. Adaptive thermogenesis describes a physiological process that can be induced by exposure to cold. Energy is hereby dissipated in mitochondria within the brown adipose tissue (BAT) and skeletal muscle in the form of heat [16]. Three categories of adaptive thermogenesis are known: cold-induced shivering thermogenesis, cold-induced non-shivering thermogenesis, and diet-induced thermogenesis [6,16]. Cold-induced non-shivering and diet-induced thermogenesis both proceed in the BAT where food energy is utilized for heat production. The uncoupling protein 1 (*Ucp1*) hereby realizes heat production by uncoupling the mitochondrial respiration from the adenosine triphosphate synthesis [6,17]. Importantly, only heat that is produced by cold-induced non-shivering thermogenesis in BAT is utilized for the maintenance of body temperature [16]. A key mechanism for cool adaptation in mice entails an increased turnover in adipose depots, in particular the activation of BAT [5,17].

*Ucp1* mRNA expression in the interscapular BAT of mice after restraint in a metabolic cage can be analyzed to investigate whether *Ucp1* mRNA is upregulated in the context of cold-induced non-shivering thermogenesis.

Taking the mouse thermal physiology into account is crucial since mice are homoiothermal animals and thermogenesis for the maintenance of body temperature is therefore essential to ensure physiological function [6]. Mice are small mammals possessing a large surface area in relation to their small volume, which predisposes them to hypothermia due to increased heat loss [18,19].

Thermoneutrality depends on many factors including body size, body composition, fur condition, energy expenditure, age, and sex [20]. To our knowledge, a systematic analysis of cold stress and the associated impact on the animal welfare of mice during metabolic cage housing has not been conducted so far. Therefore, our study aimed to minimize cold stress and at the same time to improve the welfare of laboratory mice during restraint in metabolic cages by developing an innovative metabolic cage (IMC) and by directly comparing the IMC with one commercially available metabolic cage from Tecniplast GmbH (TMC) and a control cage. Our data show that the IMC is a more effective tool compared to the TMC for ensuring mouse welfare to the greatest extent possible mainly realized by a reduction in cold stress. Here, we investigated body weight and composition, food intake, the fur condition of mice, and fecal corticosterone metabolites (CMs) to assess the degree of welfare reduction in the TMC versus the IMC. Additionally, thermal images of mice were taken and mRNA expression levels of *Ucp1* in BAT were examined to directly assess the extent of cold stress in both types of metabolic cages.

## 2. Materials and Methods

### 2.1. Ethical Statement

The animal experiment was approved by the corresponding authority in Brandenburg, Germany (Landesamt für Arbeitsschutz, Verbraucherschutz und Gesundheit, permit number: 2347-14-2019). All interventions were performed in compliance with the German Animal Welfare Act. The animal experiment was pre-registered as a non-technical project summary (NTP) on https://www.animaltestinfo.de/ (NTP-ID: 00029026-1-3) (accessed on 1 March 2019). Mice were handled and housed according to recommendations and guidelines of the Federation of European Laboratory Animal Science Associations (FELASA) [21,22] and the Society of Laboratory Animal Science (GV-SOLAS) [23].

### 2.2. Animals and Housing Conditions

C57BL/6J mice, exemplary of other mouse strains, were selected for experiments since this inbred mouse strain is most frequently used in biomedical research. Female and male mice were utilized for experiments from in-house breeding in the central laboratory animal husbandry at the Max Rubner Laboratory of the German Institute of Human Nutrition, Germany. Mice were bred and held under specified pathogen-free conditions and all experiments were also conducted under specified pathogen-free hygiene standards. Before the start of experiments, females and males were housed separately in groups of five together with their sibling animals in conventional type III cages (800 cm^2^, EHRET GmbH Life Science Solutions, Freiburg, Germany). After the first 24 h of single-housing in control cages/Tecniplast metabolic cage (TMC)/innovative metabolic cage (IMC), male mice were further single-housed in type II open cages (350 cm^2^, Tecniplast GmbH, Hohenpeißenberg, Germany). This was necessary due to their aggressive behavior and relates to the 6 d recovery periods between both 24 h single-housing sessions in control cages/TMC/IMC. Since open cage systems were used, olfactory as well as visual contact between mice was maintained at all times.

Home cages and control cages contained aspen wood bedding material (grain size: 2–5 mm, height: 1.5 mm, ssniff Spezialdiäten GmbH, Soest, Germany) and standardized enrichment including the following: 1 cotton nestlet (5 cm × 5 cm, ZOONLAB GmbH, Castrop-Rauxel, Germany), 2 cellulose tissues (green, H3-towel system classic, 23 cm × 24.8 cm, Essity Professional Hygiene Germany GmbH, Mannheim, Germany), 1 aspen wood gnawing bar (100 mm × 20 mm × 20 mm, ssniff Spezialdiäten GmbH, Soest, Germany), and 1 cardboard house (16 cm × 12 cm × 8 cm, LBS Biotechnology, London, UK). Standard conditions were maintained in the animal husbandry as follows: room temperature 23 °C (1), relative humidity 50% (10), light:dark cycle 12:12 h of artificial light (lights on, 06:00 a.m. to 06:00 p.m.; dusk–dawn system, 06:00–06:30 p.m. and 06:00–06:30 a.m.). A pelleted diet was fed to the mice (rat/mouse maintenance, V 1534-300, ssniff Spezialdiäten GmbH, Soest, Germany). Acidified water (pH 2.5–3.0) and autoclaved food pellets were provided ad libitum. Two animal caretakers and two experimenters were assigned to handle the mice during breeding and experiment in order to minimize stress reactions.

### 2.3. Description of Metabolic Cages

Metabolic cages are used for the continuous collection and clean separation of fecal samples and urine. For this purpose, the cage ground represents a grid with a funnel system underneath guiding samples into collecting vessels. We conducted a comparative study by investigating two different metabolic cage types and directly compared these two metabolic cage systems with a control cage. One cage was constructed in the research workshop at the German Institute of Human Nutrition on the basis of the metabolic cage model constructed by Hatteras Instruments, Inc. (model: MMC100, obese design; Grantsboro, NC, USA) and is designated as the innovative metabolic cage (IMC). The other metabolic cage type is commercially available at Tecniplast GmbH (metabolic cages for individual mice, 3600M021; Hohenpeißenberg, Germany) and is designated as the Tecniplast metabolic cage (TMC). The details of both cages are shown in Appendix A. Substantial refinement measures were incorporated into the cage construction of the IMC and the improvements for designing the IMC model are as follows: (1) The grid construction induces pain in the forepaws and hind paws of mice due to the sensitization of palmar and plantar nerves. Therefore, the mesh size of the stainless steel grid was reduced from 0.6 cm × 3.5 cm (TMC) to 0.4 cm × 0.4 cm (IMC) with the aim of optimizing the weight and pressure distribution on the paws. (2) A resting platform (diameter: 2.5 cm) was added centrally on top of the cage grid. This platform, made out of plastic material (polyvinyl chloride), should provide the possibility for resting paws during restraint in IMC. Additionally, the plastic material supports heat conductance, in contrast to the metal grid, which dissipates heat. (3) The total cage volume of the IMC was reduced to 1.2 dm^3^ (used cage space of mice from grid to lid) in order to support mice with the heating up of the metabolic cage due to the reduction in air volume. The TMC possesses a larger cage volume of 3.1 dm^3^. The IMC is made of plexiglass. (4) The water and food consumption of mice can be monitored in metabolic cages, but water and food are poorly accessible for mice in the TMC since animals need to enter the prechambers (see Appendix A). Therefore, the supply of food and water ad libitum cannot be guaranteed. Access to the food and water supply is facilitated in IMC by an angled food hopper and water supply, omitting the use of prechambers.

### 2.4. Study Design

A total number of 25 female and 25 male C57BL/6J mice at the age of 66–73 d were used. Mice were randomly assigned by one experimenter to the control group (n = 5 females and n = 5 males), TMC group (n = 10 females and n = 10 males), and IMC group (n = 10 females and n = 10 males). Only one experimenter was aware of the group allocation during the conduct of the experiment, outcome assessment, and data analysis. The control group was housed in type II open cages (350 cm^2^, Tecniplast GmbH, Hohenpeißenberg, Germany) to investigate the effect of single-housing itself.

Baseline values of the fur score were determined in order to assess the animal’s welfare before the experiments started (see Appendix A). 

Thereafter, both sexes were returned to their home cages with familiar group constellations until the start of experiments. After a 6 d resting period, mice were transferred into either the control cage, TMC, or IMC. Shortly before the transfer, the body weight and body composition of each mouse were assessed. Mice were single-housed for 24 h. Thermal images of each mouse were taken with an infrared camera at the beginning and the end of the 24 h time period for evaluating cold stress. The fur score of each mouse was assessed again. Body weight and body composition data of each mouse were gathered again afterwards. Thereafter, females were returned to their home cages in groups of five animals while males were single-housed until the end of the experiments. Metabolic cage data were gathered including the food intake and the total fecal output. After the first restraint into different cage systems, a 6 d resting period was provided followed by a second restraint according to the same experimental workflow. At the end of the second restraint, the animals were killed via inhalative isoflurane narcosis and exsanguination. The order of treatments was scheduled so that invasive methods (body weight and composition) were performed immediately before and after the stress-inducing metabolic cage restraint and otherwise solely non-invasive methods were applied (thermal imaging, fur score).

### 2.5. Humane Endpoints: Termination Criteria

Animal health was monitored on a daily basis during the two-week trial. Death was not accepted as an endpoint in the present study design at any time. None of the utilized mice for conducting the described experiment reached our predefined termination criteria. The respective mice would have been excluded from the experiment at the earliest possible point in time, i.e., the animal would have been preemptively euthanized without pain. The aim was to ensure that the data generated by this animal up to that point remained usable. Thus, euthanasia would have been performed well before a moribund state was reached. A score of 2 should not be reached (see Table 1). This refers to the time frame of the 6 d recovery period after the first 24 h of single-housing in control cages/TMC/IMC, i.e., when the evaluated parameters of the mice have not stabilized by the end of this recovery period. Accordingly, no second single-housing session would have taken place.

### 2.6. Food Intake

Food intake was manually measured over a period of 24 h during restraint in both metabolic cage types. The food consumption of mice held in control cages for 24 h was not monitored. Group-housed (female) and single-housed (male) mice in home cages during the 6 d resting period were also not included in these analyses. Food pellets were placed into the food hoppers of both metabolic cage types and the food hopper was weighed before (empty weight) and after filling with food. Additionally, shredded food pellets from all cage equipment were collected in a weighing dish and their weight was factored into the calculations to avoid overestimation of food intake.

### 2.7. Body Weight and Body Composition

The body weight of each mouse was assessed by transferring mice to a metal bowl which was positioned on top of a scale and the mean value calculation for the body weight determination of moving mice was selected (F button; PLJ, KERN & SOHN GmbH, Balingen, Germany). Body composition, including the fat mass and lean mass of mice, was assessed by nuclear magnetic resonance (Echo-MRI^TM^, Zinsser Analytic GmbH, Eschborn, Germany). This measurement represents a non-invasive technique since mice do not need to undergo anesthesia and are transferred and fixated into a 6 cm ⌀ plexiglass tube for this purpose. The measuring time is approx. 1.5 min. Mice were placed in the tube directly from the balance after body weight measurement. After body composition measurement, the tube containing the mouse was positioned in the home cage in order to release the animal. In total, the body weight and body composition of each mouse were determined four times between 08:00 a.m. and 12:00 a.m. during the experiment: before the first restraint (home cage), after the first restraint (control/TMC/IMC), before the second restraint (home cage), and after second restraint (control/TMC/IMC). Body weight and body composition changes were calculated as percentages based on body weight, fat, and lean mass assessed shortly before and after the 24 h restraint.

### 2.8. Determination of Cage and Body Surface Temperature with Thermal Imaging

In order to assess cold stress, thermal images of mice within the three tested cage types were taken with a thermal imaging camera (E50, FLIR, Frankfurt am Main, Germany). Thermal images were taken without cage lids between 08:00 a.m. and 10:00 a.m. before and after the 24 h single-housing. The thermal imaging camera settings were defined as follows: temperature scale, 20–38 °C; color palette, lava; atmospheric temperature, 23 °C; object distance, 1 m; relative humidity, 50%; emittance, 0.95; and reflected apparent temperature, 20 °C. The infrared camera assessed three points within each picture. The middle point (2) was located in the area from the interscapular to the thoracolumbar region of each mouse, measuring the body surface temperature, and the outer points (1 and 3) assessed the cage temperature. The cage temperatures of points 1 and 3 were averaged if the difference between both points was not > 0.6 °C.

### 2.9. Expression of Uncoupling Protein 1 in Brown Adipose Tissue

Total RNA from brown adipose tissue (BAT) was isolated by phenol–chloroform extraction with Invitrogen^TM^ TRIzol^TM^ Reagent (see TRIzol Reagent User Guide: Isolate RNA, Doc. Part No. 15596026.PPS, Pub. No. MAN0001271, Rev. B.0, according to manufacturer’s instructions, Thermo Fisher Scientific, Waltham, MA, USA). BAT was pulverized under liquid nitrogen with a mortar. An amount of 20 mg of BAT was utilized for RNA isolation. For additional purification of RNA isolates, DNase from *E. coli* cells was added. An amount of 8 µg of RNA isolate was added to 2 µL of DNase (1 u/µL) and the reaction mixture was filled up to 30 µL with reaction buffer (DNase I, RNase-free Kit, Thermofisher Scientific, Waltham, MA, USA). The mixture was incubated for 30 min (600 rpm, 37 °C). An amount of 1 µL of EDTA (50 mM) was added and incubated for 10 min (600 rpm, 65 °C). 

Isolated RNA samples were stored at −80 °C. cDNA was synthesized from 2 µg of purified RNA (Revertaid^TM^ First Strand cDNA Synthesis Kit, Thermofisher Scientific, Waltham, USA). An amount of 1 µL of oligo (dT)_18_ primers (100 pmol) was added, the reaction mixture was filled up to 11 µL with nuclease-free H_2_O, and samples were incubated for 5 min (65 °C) followed by the cooling of samples to 4 °C. An amount of 4 µL of 5× reaction buffer, 1 µL of nuclease-free H_2_O, 2 µL of dNTP mix (10 mM), and 1 µL of RevertAid M-MuL V Reverse Transcriptase (200 U) were added to each sample. Samples were incubated for 60 min at 45 °C with subsequent heating to 70 °C for 5 min. cDNA synthesis was terminated with the cooling of samples to 4 °C and cDNA was stored at −20 °C until the performance of RT-PCR. RT-PCR for the quantification of each transcript was carried out in duplicate in a reaction mixture of SYBR Green/Fluorescin qPCR Master Mix (Thermo Fisher Scientific, Waltham, United States), 10 µM forward and reverse oligonucleotides (Eurofins Genomics, Ebersberg, Germany), and 4.5 µL cDNA in a total volume of 10 µL. RT-PCR was performed in a LightCycler^®^ (Roche Deutschland Holding GmbH, Mannheim, Germany) with an initial enzyme activation step at 95 °C for 10 min, followed by 42 cycles of denaturation at 95 °C for 15 s, annealing at 58 °C for 15 s, and elongationat 72 °C for 15 s. Relative gene expression of *Ucp1* (fw: TGGTGAACCCGACAACTTCC, rv: GGCCTTCACCTTGGATCTGAA, 141 bp) was normalized to the expression levels of two reference genes (Hypoxanthine-guanine-phosphoribosyltransferase 1—fw: TGGATACAGGCCAGACTTTGTT, rv: CAGATTCAACTTGCGCTCATC, 162 bp; β-actin—fw: CCAGCCTTCCTTCTTGGGTAT, rv: GGGTGTAAAACGCAGCTCAG, 374 bp) by applying the following formula:(1)$$Fold\;induction=2^{\left(a-b\right)\;gene\;of\;interest}/2^{\left(a-b\right)\;reference\;genes}$$

Parameter a represents the arithmetic mean of all Ct-values from the BAT samples of the control group and parameter b is the Ct-value of every single BAT sample assigned to either the TMC or IMC group. Since two reference genes were applied for normalization, the geometric mean of the difference (*a* − *b*) of each reference gene was calculated [24].

### 2.10. Assessment of Grooming State by Using the Fur Score

The fur score represents an objective tool to determine the grooming state of fur in order to draw conclusions on mouse welfare and corresponding stress levels. The fur quality of mice was assessed between 8:00 a.m. and 10:00 a.m. Two observers assigned scores at each acquisition date independently and both scores were averaged for each time. The fur score comprises a four-degree scale according to Mineur et al. [25].

### 2.11. Determination of Fecal Corticosterone Metabolites

Fecal samples were collected with forceps from the bedding of the control cages, collection vessels of both metabolic cages, and the close-meshed IMC grid. After determination of the fresh weight, feces were stored at −80 °C until analysis. The hypothalamic pituitary adrenal axis activity of the mice was assessed by measuring immunoreactive fecal corticosterone metabolite (CM) concentrations. This technique represents a non-invasive alternative for assessing stress hormone concentrations in mice [26,27]. Quantification of fecal CM was performed in duplicates by means of a self-developed enzyme immunoassay (EIA) system. The collected fecal samples were analyzed for immunoreactive CM using a 5α-pregnane-3β,11β,21-triol-20-one EIA. Details regarding the development, biochemical characteristics, and physiological validation of this assay are described by Touma and colleagues [26,27]. Before EIA analysis, the fecal samples were dried for about 3 h in an oven at 80 °C (UN30, SingleDISPLAY, Memmert GmbH + Co. KG, Schwabach, Germany), crushed to powder with a mortar (Technical Ceramics, Waldkraiburg, Germany), and aliquots of 0.05 g were extracted by shaking for 30 min (orbital: 60 rpm, 10 s; reciprocal: 90°, 30 s; vibro/pause: 1°, 3 s; (Multi-1 PRS-26, Starlab International GmbH, Hamburg, Germany)) with 1 mL of 80% methanol (HiPerSolv CHROMANORM, VWR International GmbH, Darmstadt, Germany). Fecal matter was separated by centrifugation for 10 min (4000× *g*) and the supernatant was stored at −20 °C until EIA analysis. Details of the EIA are specified in the publications of Touma et al. [26,27]. 

### 2.12. Statistical Analyses

Sample size calculation was performed based on body weight change during restraint in the metabolic cage (power: 0.83; sample size: 12; alpha: 0.05; difference in means: 5.0; standard deviation: 4.0). Plots were generated by GraphPad Prism version 6 (Graphstats Technologies Private Limited, Karnataka, India) and display the mean (standard deviation). Statistical analyses were performed using IBM SPSS Statistics version 20 (IBM Corporation, Armonk, NY, USA). Differences between the two groups were studied by application of an independent sample t-test (normal distribution of data, two-tailed) or the Mann–Whitney U test. For statistical analyses of differences between the three groups, a one-way analysis of variance (ANOVA) was utilized with the following post hoc tests: Tukey’s HSD (normal distribution, equal variances) and Dunnett T3 (no equal variances). Statistical significance between tested groups was accepted when *p* < 0.05.

## 3. Results

### 3.1. Changes in Body Weight, Body Composition, and Food Intake Are Dependent on the Metabolic Cage Type

The phenotypic characterization of mice single-housed for a period of 24 h indicated that changes in body weight, fat, and lean mass as well as food intake are dependent on which metabolic cage type was applied (see Figure 1A–D and Figure 2A–D). A one-way ANOVA was performed to compare the effect of cage type on body weight change for both restraints and sexes separately. For females, there was a statistically significant difference in body weight change during both restraints between at least two cage types (1st restraint: F(2,22) = 11.024, *p* < 0.001; 2nd restraint: F(2,22) = 24.498, *p* < 0.001). Dunnett’s T3 test for multiple comparisons found that the mean value of body weight change for females during both restraints was significantly different between the TMC (1st restraint: −13.2% (6.0), 2nd restraint: -15.0% (4.7)) and control mice (1st restraint: −3.8% (1.7), *p* = 0.002, 95% C.I. = 3.706, 15.021; 2nd restraint: −4.8% (0.6), *p* < 0.001, 95% C.I. = 5.908, 14.600), but also between the TMC and IMC mice (1st restraint: −6.9% (1.2), *p* = 0.028, 95% C.I. = 0.698, 11.703; 2nd restraint: −6.9% (1.3), *p* = 0.001, 95% C.I. = 3.790, 12.491; see Figure 1A). Concerning males, a one-way ANOVA analysis showed that there was a statistically significant difference in body weight change during both restraints between at least two of the tested cage types (1st restraint: F(2,22) = 7.798, *p* = 0.003; 2nd restraint: F(2,22) = 34.239, *p* < 0.001). Tukey’s HSD test for multiple comparisons found that the body weight change in males during both restraints was more pronounced in the TMC (1st restraint: −13.1% (3.4), 2nd restraint: −14.9% (4.0)) compared to controls (1st restraint: −6.3% (4.5), *p* = 0.006, 95% C.I. = 1.831, 11.614; 2nd restraint: −3.4% (0.5), *p* < 0.001, 95% C.I. = 7.424, 15.625) and in the TMC compared to IMC (1st restraint: −8.1% (3.2), *p* = 0.012, 95% C.I. = 1.030, 9.017; 2nd restraint: −5.8% (2.4), *p* < 0.001, 95% C.I. = 5.744, 12.440; see Figure 1B). No significant difference in body weight change for male mice between control and IMC was detected (1st restraint: *p* = 0.663, 2nd restraint: *p* = 0.315; see Figure 1B). After 6 d of resting, all mice recovered and reached a body weight that was comparable to the initial body weight. This applies to the period between the two restraints. Females were grouped together in their previous group constellations, but males were single-housed until the end of the experiment. Food intake is expressed per body weight to account for the significant change in body weight during metabolic cage housing (see Figure 1C,D). After the first restraint for both sexes, the independent sample t-test indicated no statistically significant difference in food intake between the TMC (females: 0.17 g/g body weight (0.12); males: 0.12 g/g body weight (0.07)) and IMC (females: 0.18 g/g body weight (0.03), t(10.048) = −0.315, *p* = 0.759; males: 0.17 g/g body weight (0.06), t(18) = −1.817, *p* = 0.086; see Figure 1C,D). Females showed a tendency to consume less food during the second restraint in the TMC (0.11 g/g body weight (0.12)) than in IMC (0.19 g/g body weight (0.02), t(9.514) = −2.177, *p* = 0.056; see Figure 1C). Food intake for male mice during the second restraint was significantly higher in IMC (0.17 g/g body weight (0.03)) compared to the TMC (0.07 g/g body weight (0.04), t(18) = −5.479, *p* < 0.001; see Figure 1D).

A one-way ANOVA was performed to compare the effect of cage type on lean and fat mass changes for both restraints and sexes. For female and male mice, there was a statistically significant difference in lean mass change during both restraints between at least two cage types (females—1st restraint: F(2,22) = 14.342, *p* < 0.001; 2nd restraint: F(2,22) = 24.609, *p* < 0.001; males—1st restraint: F(2,22) = 7.034, *p* = 0.004; 2nd restraint: F(2,22) = 36.396, *p* < 0.001). For females, Dunnett’s T3 test for multiple comparisons revealed that lean mass change during both restraints in the TMC (1st restraint: −13.0% (4.4), 2nd restraint: −14.5% (3.5)) was significantly higher compared to controls (1st restraint: −5.3% (1.4), *p* = 0.01, 95% C.I. = −11.826, −3.459; 2nd restraint: −5.9% (1.7), *p* < 0.001, 95% C.I. = −12.245, −4.949) and IMC (1st restraint: −7.5% (1.0), *p* = 0.009, 95% C.I. = −9.527, −1.471; 2nd restraint: −7.9% (1.7), *p* < 0.001, 95% C.I. = −9.919, −3.256; see Figure 2A). Concerning male mice during both restraints, Tukey’s HSD test for multiple comparisons showed that the mean value of lean mass change was significantly different between the TMC (1st restraint: -12.3% (3.0), 2nd restraint: −12.9% (2.4)) and controls (1st restraint: −6.0% (4.2), *p* = 0.009, 95% C.I. = −11.064, −1.448; 2nd restraint: −4.4% (0.9), *p* < 0.001, 95% C.I. = −11.531, −5.338) as well as the TMC and IMC (1st restraint: −7.6% (3.6), *p* = 0.017, 95% C.I. = −8.635, −0.784; 2nd restraint: −5.4% (2.5), *p* < 0.001, 95% C.I. = −10.021, −4.964; see Figure 2B).

Importantly, no significant differences between lean mass change in control cages and IMC were detected for both sexes and restraints (females—1st restraint: *p* = 0.057, 2nd restraint: *p* = 0.166; males—1st restraint: *p* = 0.702, 2nd restraint: *p* = 0.728; see Figure 2A,B). For female mice during the first restraint, no statistically significant difference in fat mass change between the three cage types (control: −20.8% (9.6), TMC: −34.8% (18.6), and IMC: −21.6% (5.5)) was detected (F(2,22) = 3.227, *p* = 0.059; see Figure 2C). However, during the second restraint of female mice, there was a statistically significant difference in fat mass change between at least two cage types (F(2,22) = 9.798, *p* = 0.001). Dunnett’s T3 test for multiple comparisons found that females lost significantly more fat mass in the TMC (−44.8% (18.0)) compared to IMC (−20.6% (5.1), *p* = 0.006, 95% C.I. = −40.820, −7.663; see Figure 2C). Concerning male mice, a statistically significant difference in fat mass change was detected between at least two cage types during both restraints (1st restraint: F(2,22) = 9.221, *p* = 0.001; 2nd restraint: F(2,22) = 21.128, *p* < 0.001). Tukey’s HSD test for multiple comparisons found that the fat mass change during both restraints was significantly higher in the TMC (1st restraint: −45.4% (17.0), 2nd restraint: −58.3% (15.7)) in contrast to controls (1st restraint: −23.8% (9.7), *p* = 0.031, 95% C.I. = −41.373, −1.824; 2nd restraint: −21.1% (5.9), *p* < 0.001, 95% C.I. = −55.595, −18.864) and IMC (1st restraint: −18.8% (13.2), *p* = 0.001, 95% C.I. = −42.728, −10.437; 2nd restraint: −23.9% (13.1), *p* < 0.001, 95% C.I. = −49.422, −19.431; see Figure 2D). As for lean mass change, fat mass change during both restraints for both sexes was not significantly different between IMC and controls (females—1st restraint: *p* = 0.998, 2nd restraint: *p* = 0.116; males—1st restraint: *p* = 0.804, 2nd restraint: *p* = 0.922; see Figure 2C,D). In summary, it can be stated that both body weight and body composition changes were more pronounced after 24 h single-housing in the TMC compared to controls and IMC. A tendency or trend for decreased food intake in the TMC compared to the IMC was concomitant.

**Figure 2 animals-13-02866-f002:**
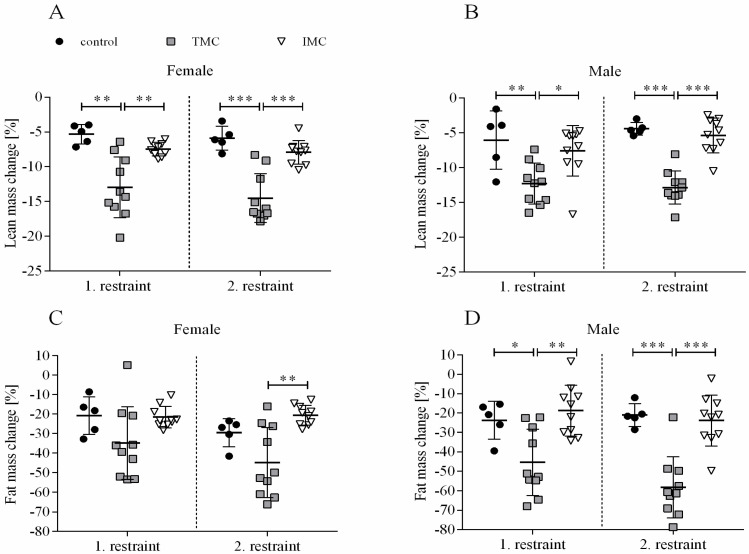
Lean mass and fat mass changes in female (**A,C**) and male (**B,D**) mice. Female and male C57BL/6J mice were single-housed in one of the three cage types (control cage, Tecniplast metabolic cage, or innovative metabolic cage). The restraint was limited to 24 h and solely repeated once, totaling two restraints. Control, control cage (n = 5); TMC, Tecniplast metabolic cage (n = 10); IMC, innovative metabolic cage (n = 10). All data are presented as means (standard deviation). Differences between TMC and IMC, TMC and control, and IMC and control were calculated by one-way ANOVA. * *p* < 0.05, ** *p* < 0.01, *** *p* < 0.001.

### 3.2. The Cold Stress for Mice Is Attenuated during Restraint in the Innovative Metabolic Cage

Besides the phenotypic characterization, the cold stress of the mice during metabolic cage restraint was examined in more detail by assessing the cage and body surface temperature with thermal imaging (see Figure 3A–C). A one-way ANOVA was performed to compare the effect of cage type on cage and body surface temperature. For both sexes and restraints, there was a statistically significant difference in cage temperature between at least two cage types (females—1st restraint: F(2,22) = 88.574, *p* < 0.001; 2nd restraint: F(2,22) = 40.545, *p* < 0.001; males—1st restraint: F(2,21) = 7.636, *p* = 0.003; 2nd restraint: F(2,22) = 22.988, *p* < 0.001). At the end of both restraints, Tukey’s HSD test for multiple comparisons found that the cage temperature for female mice was significantly higher in the IMC (1st restraint: 23.7 °C (0.3), 2nd restraint: 23.5 °C (0.2)) compared to the control (1st restraint: 22.4 °C (0.2), *p* < 0.001, 95% C.I. = 0.979, 1.641; 2nd restraint: 22.5°C (0.3), *p* < 0.001, 95% C.I. = 0.601, 1.339; see Figure 3D). In addition, the cage temperature for female mice housed in the TMC (1st restraint: 22.4 °C (0.3), 2nd restraint: 22.5 °C (0.3)) was significantly reduced in contrast to the IMC (1st restraint: *p* < 0.001, 95% C.I. = −1.575, −1.035; 2nd restraint: *p* < 0.001, 95% C.I. = −1.297, −0.693; see Figure 3D). At the end of the first restraint, Tukey’s HSD test revealed that for male mice the cage temperature in the TMC (22.5 °C (0.2)) was significantly reduced compared to controls (23.1 °C (0.4), *p* = 0.049, 95% C.I. = −1.227, −0.027; see Figure 3D). At the end of the second restraint, Dunnett’s T3 test revealed that IMC temperatures (23.5 °C (0.4)) for males were significantly higher compared to controls (22.2 °C (0.8), *p* = 0.045, 95% C.I. = 0.036, 2.464; see Figure 3D). After both restraints, the cage temperature of males kept in IMC (1st restraint: 23.3 °C (0.6), 2nd restraint: 23.5 °C (0.4)) was significantly higher compared to the TMC (1st restraint: 22.5 °C (0.2), *p* = 0.003, 95% C.I. = 0.248, 1.275; 2nd restraint: 22.4 °C (0.2), *p* < 0.001, 95% C.I. = 0.742, 1.448; see Figure 3D). The higher cage temperatures measured in the IMC compared to the control cage and/or TMC clearly indicate a positive effect on the thermal regulation of the mice due to the volume reduction in the IMC construction. Concerning the body surface temperature for both sexes at the end of the first restraint, one-way ANOVA analyses revealed that there was no statistically significant difference between cage types (females—control: 31.2 °C (0.8), TMC: 30.1 °C (0.6), IMC: 30.9 °C (1.5); F(2,22) = 2.381, *p* = 0.116; males—control: 31.4 °C (0.6), TMC: 30.3 °C (0.9), IMC: 31.3 °C (1.2); F(2,21) = 3.162, *p* = 0.063; see Figure 3E). At the end of the second restraint for both sexes, there was a statistically significant difference in body surface temperatures between cage types (females: F(2,22) = 15.351, *p* < 0.001; males: F(2,22) = 7.429, *p* = 0.003). Tukey’s HSD test found that the mean value of body surface temperature at the end of the second restraint for female mice was significantly higher in IMC (31.7 °C (0.8)) compared to controls (30.3 °C (1.0), *p* = 0.044, 95% C.I. = 0.337, 2.626; see Figure 3E). Concerning male mice at the end of the second restraint, Tukey’s HSD test revealed that the body surface temperature was significantly reduced in the TMC (30.0 °C (0.8)) compared to the control (31.1 °C (0.9), *p* = 0.044, 95% C.I. = −2.254, −0.026; see Figure 3E). For both sexes at the end of the second restraint, the body surface temperature in the IMC (females: 31.7 °C (0.8), males: 31.3 °C (0.8)) was significantly increased compared to the TMC (females: 29.3 °C (1.1), *p* < 0.001, 95% C.I. = 1.272, 3.389; males: 30.0 °C (0.8), *p* = 0.004, 95% C.I. = −2.240, −0.430; see Figure 3E). As for the cage temperature, an increase in body surface temperature was assessed at the end of the second restraint in the IMC compared to the TMC. This can be attributed to the improvement in the IMC construction actively supporting the mice in thermoregulation when they are unable to build a nest in the metabolic cage. Next, we investigated uncoupling protein 1 (*Ucp1*) expression in brown adipose tissue (BAT), because we aimed to investigate whether *Ucp1* mRNA is upregulated in the context of cold-induced thermogenesis, i.e., metabolic cage restraint [28,29,30]. A one-way ANOVA revealed that there was no statistically significant difference in the mRNA expression of *Ucp1* in BAT of female mice relative to the control group among all three tested cage types (control: 1.1 (0.3); TMC: 1.2 (0.4); IMC: 1.4 (0.4); F(2,21) = 1.274, *p* = 0.300; see Figure 3F). Concerning male mice, a one-way ANOVA found that there was a statistically significant difference in *Ucp1* mRNA expression in BAT between at least two tested cage types (F(2,22) = 4.483, *p* = 0.023), but Dunnett’s T3 test for multiple comparisons did not reveal a statistically significant difference between the three tested cage types (control: 1.2 (0.9); TMC: 0.6 (0.3); IMC: 0.5 (0.2); TMC vs. control: *p* = 0.406; IMC vs. control: *p* = 0.370; TMC vs. IMC: *p* = 0.988; see Figure 3F).

The fur score is an effective tool to objectively assess the status of animal welfare. An increase in the fur score is associated with a reduction in grooming resulting in a fluffy fur appearance. The deterioration in the fur condition is paralleled by a higher stress level in mice and could furthermore serve to combat cold stress by raising the fur hairs and promoting heat insulation. In terms of a multifactorial approach for the severity assessment of metabolic cage restraint, the fur score was applied to additionally address the cold stress of mice using another approach (clinical scoring) [31]. The quantitative parameters, cage and body surface temperature, as well as mRNA expression of *Ucp1* in BAT describe the physiological state of the mouse. The black fur color and generally smooth fur state of C57BL/6J mice often resulted in equal fur scores given by both independent scorers (standard deviation = 0.00) and a clear sex difference concerning the basal fur score was not apparent (fur score = 1.00, see Table 2). The Mann–Whitney U test indicated that for female and male mice the fur scores after both restraints in the TMC (females—both restraints: 2.00 (0.00); males—1st restraint: 2.75 (0.26), 2nd restraint: 2.50 (0.53)) were significantly increased compared to controls (females—1st and 2nd restraint: 1.00 (0.00), U(10,5) = 0.000, *p* = 0.001; males—1st and 2nd restraint: 1.00 (0.00), U(10,5) = 0.000, *p* = 0.001; see Table 2). Also concerning both sexes and restraints, the fur scores in the IMC (females and males—both restraints: 2.00 (0.00)) were significantly increased compared to controls (females and males—both restraints: 1.00 (0.00), U(10,5) = 0.000, *p* = 0.001; see Table 2). After the first restraint, the fur scores for male mice housed in the TMC (2.75 (0.26)) were significantly higher compared to the fur scores of males in the IMC (2.00 (0.00), U(10,10) = 0.000, *p* < 0.001, see Table 2). For female mice after the first and second restraint, there was no statistically significant difference in fur scores between the TMC (both restraints: 2.00 (0.00)) and IMC (both restraints: 2.00 (0.00), U(10,10) = 50.000, *p* = 1.000; see Table 2).

### 3.3. Excretion of Corticosterone Metabolites in the Feces Is Decreased by Restraining Mice in the Innovative Metabolic Cage

For monitoring stress hormone levels during metabolic cage housing, corticosterone metabolites (CM) were quantified in fecal samples collected during each restraint. A one-way ANOVA was performed to compare the effect of cage type on the excretion of CM in the feces. Significant differences in fecal CM excretion between at least two of the tested cage types were revealed for both sexes and restraints (females—1st restraint: F(2,22) = 4.348, *p* = 0.026; 2nd restraint: F(2,22) = 5.014, *p* = 0.016; males—1st restraint: F(2,22) = 5.837, *p* = 0.009; 2nd restraint: F(2,22) = 7.459, *p* = 0.003). For the females during the first restraint, Tukey’s HSD test for multiple comparisons found that the mean value of fecal CM for the TMC (1375.7 ng/g dry weight (DW) (585.7)) was significantly higher compared to controls (704.6 ng/g DW (226.5), *p* = 0.020, 95% C.I. = 98.880, 1243.340; see Figure 4A). Dunnett’s T3 test revealed the same trend for females during the second restraint (TMC: 2098.2 ng/g DW (1067.2); control: 940.8 ng/g DW (254.7); *p* = 0.023, 95% C.I. = 163.555, 2151.065; see Figure 4A). For females during both restraints, fecal CM levels in IMC (1st restraint: 1130.7 ng/g DW (238.6); 2nd restraint: 1242.3 ng/g DW (493.6)) and control cages (1st restraint: 704.6 ng/g DW (226.5); 2nd restraint: 940.8 ng/g DW (254.7)) were not significantly different from each other (1st restraint: *p* = 0.171; 2nd restraint: *p* = 0.355; see Figure 4A). As for the females, for males, Dunnett’s T3 test found that fecal CM excretion during both restraints was significantly elevated in the TMC (1st restraint: 1029.9 ng/g DW (427.8); 2nd restraint: 1068.9 ng/g DW (400.9)) compared to control cage housing (1st restraint: 504.4 ng/g DW (54.6), *p* = 0.010, 95% C.I. = 133.551, 917.449; 2nd restraint: 537.1 ng/g DW (117.9), *p* = 0.006, 95% C.I. = 208.827, 1130.173; see Figure 4B). In comparison to control cage housing (504.4 ng/g DW (54.6)) during the first restraint, male mice housed in IMC showed significantly elevated fecal CM levels (1080.9 ng/g DW (269.6), *p* < 0.001, 95% C.I. = 327.288, 825.692), but not during the second restraint (control: 537.1 ng/g DW (117.9); IMC: 753.3 ng/g DW (214.3); *p* = 0.073; see Figure 4B). Separating the fecal CM concentrations into “well-groomed” mice (fur score = 1; fecal CM: females—897.7 ng/g DW (246.3), males—520.7 ng/g DW (88.3)) and “less groomed” mice (fur score > 1; fecal CM: females—1520.9 ng/g DW (780.1), males—1017.6 ng/g DW (392.7)) resulted in a significant difference between both tested data sets (females: U(40,35) = 226.000, *p* < 0.001; males: t(68.056) = −5.527, *p* < 0.001; see Figure 4C,D). A fur score of 1 indicating a well-groomed fur was only assigned during baseline assessment and for controls. Accordingly, it can be concluded that mice possessing a less groomed fur excrete increased CM via the feces, and this is solely attributed to the metabolic cage restraint in the context of this study.

## 4. Discussion

In the interest of refining animal experiments, in this study we aimed to improve the welfare of laboratory mice during metabolic cage restraint, mainly by minimizing cold stress. Two different metabolic cage types and a control cage were compared within the framework of a multifactorial approach for severity assessment including parameters such as body weight, body composition, food intake, cage and body surface temperature, mRNA expression of uncoupling protein 1 (*Ucp1)* in brown adipose tissue (BAT), fur score, and fecal corticosterone metabolites (CM).

Specifically because of the increased cold stress of mice during metabolic cage restraint in comparison with conventional housing conditions including nesting and bedding material, group-housing, and enrichment, mice have to mobilize additional energy resources to approximate thermoneutral conditions. It is also important to note that body weight is a decisive parameter determining the lower critical temperature (TLC) of mice [4,32]. The TLC describes the lower bound of the thermoneutral zone where the required heat precisely corresponds to the maintenance of body temperature and basal metabolic rate [4]. Beyond that, the TLC for mice is further decreased if nesting and bedding material is absent and if mice are single-housed, as is the case of restraint in metabolic cages. The TLC of mice can be calculated based on e.g., body weight, indicating a strong negative relationship between TLC and body weight, or energy expenditure [4,8]. Therefore, contradictions exist in the literature with TLCs ranging from 20 to 29 °C for C57BL/6 mice that were determined at different experimental conditions [4,8,32]. Mammals can adapt to challenging environmental conditions, such as cold stress, through numerous mechanisms including “morphological, physiological, and behavioral changes” [15]. Morphological adaptation refers to the e.g., fur condition of mice, whereby the fur state reflects the animal’s well-being, and raising of the fur hair was suggested to be associated with improved thermal insulation [1]. Physiological mechanisms entail automatic responses that are involuntarily initiated in order to produce heat after the activation of thermoreceptors in case of cold exposure [15]. *Ucp1* in the mitochondria of BAT generates extra heat for body temperature maintenance. In the case of mice and rats, this specialized form of thermoregulation is particularly important, because of their high surface area-to-volume ratio [15,18]. As heat generation by the BAT requires a high amount of energy, a hypermetabolic state is induced. This could result in a decrease in body weight as well as fat deposits of mice and could further be compensated for with an increased food intake of mice. Behavioral thermoregulation in case of cold exposure includes voluntary actions such as approaching hot habitats, nest building, or social behaviors like huddling with conspecifics [15]. As already stated, mice are deprived of behavioral thermoregulation during restraint in metabolic cages since mice are single-housed, and nesting, bedding, and enrichment materials are absent.

In the present study, we thus examined the cold stress of C57BL/6J mice specifically during restraint in two different metabolic cage types versus a control cage system at a standardized room temperature of 23 °C (1). To our knowledge, the extent of cold stress in laboratory mice during metabolic cage restraint has not yet been fully characterized, especially with regard to the parameters best suited for its accurate and at the same time non-invasive analysis. With the introduction of the innovative metabolic cage (IMC), we considerably reduced the cold stress of C57BL/6J mice during metabolic cage restraint, which was mainly shown by smaller body weight change (IMC: females—1st restraint: −6.94%; 2nd restraint: −6.89%; males—1st restraint: −8.08%; 2nd restraint: −5.82% ↔ TMC: females—1st restraint: −13.2%; 2nd restraint: −15.0%; males—1st restraint: −13.1%; 2nd restraint: −14.9%), lean mass change (IMC: females—1st restraint: −7.46%; 2nd restraint: −7.16%; males—1st restraint: −7.59%; 2nd restraint: −5.37% ↔ TMC: females—1st restraint: −13.0%; 2nd restraint: −14.5%; males—1st restraint: −12.3%; 2nd restraint: −12.9%), and an increase in cage temperature (IMC: females—1st restraint: 23.7 °C; 2nd restraint: 23.5 °C; males—1st restraint: 23.3 °C; 2nd restraint: 23.5 °C ↔ TMC: females—1st restraint: 22.4 °C; 2nd restraint: 22.5 °C; males—1st restraint: 22.6 °C; 2nd restraint: 22.4 °C) compared to housing in the Tecniplast metabolic cage (TMC). 

A second 24 h restraint in both metabolic cage types was chosen in our study design because the restraint period in other studies utilizing metabolic cages extends the period of 24 h in many cases. As this study is a refinement project, we wanted to keep the metabolic cage restraint as short as possible. To cite two studies restraining mice in metabolic cages, to be precise in the TMC, as examples, Kalliokoski et al. [1] kept male BALB/c mice in the TMC for 3 weeks while Stechman et al. [33] restrained female and male C3H, BALB/c, and C57BL/6J mice in the TMC for 7 d. The Swiss expert information [34] for metabolic cage restraint further states that a 6 d recovery period for an animal experiment with the severity category “mild” should be scheduled after restraint in metabolic cages over 8 h up to 24 h. We have based the time frame of the resting period on this guideline.

First, the phenotypic characterization of mice restrained in different cage systems was performed by determining their body weight and body composition. For female and male mice during both 24 h restraints, a significantly higher body weight change was observed in the TMC compared to IMC. Male BALB/c mice lost approximately 8% of their initial body weight after a 24 h restraint in the TMC [1]. By contrast, the body weight change in male C57BL/6J mice in the present study was 5–7% higher and amounted to −13.1% and −14.9% after the first and second 24 h restraints in the TMC. Additionally, female mice were more sensitive to the IMC than male mice since their body weight change in the IMC was significantly higher compared to control, but for males, body weight changes in the IMC and control cages were comparable.

To put the body weight change into context, we have referenced the food intake during metabolic cage housing to the body weight directly after each restraint. For both sexes, the tendency of a higher food intake during the second restraint in the IMC could be shown while male mice consumed significantly more food in the IMC in comparison to the TMC. These findings confirm our initial objective to reduce body weight change in mice in the IMC due to a higher food intake achieved by improved accessibility to the food hoppers (see detailed information in Materials and Methods: description of metabolic cages). Hereby, the process of diet-induced thermogenesis could be stimulated while increasing the food intake in order to produce heat, and the maintenance of body temperature is excluded in this type of thermogenesis [16]. Sǩop et al. demonstrated in their study that C57BL/6J mice ingested more food at lower ambient temperatures of 17–22 °C, which is comparable with our study, and less food at higher ambient temperatures of 25–34 °C [35]. It is likely that pelleted chow is less suitable for the TMC since stuck food pellets can block the supply of further pellets, which can be attributed to the fact that the food hoppers are mounted horizontally and not at an angle as in the IMC. In this case, mice are unable to meet their energy demand for the maintenance of metabolic processes at low ambient temperatures leading to the pronounced body weight loss we have observed. Kalliokoski et al. restrained male BALB/c mice in the TMC for 21 d with a powdered diet provided ad libitum [1]. Food consumption ranged from approximately 0.15 g/g body weight to 0.25 g/g body weight during TMC restraint, while male C57BL/6J mice in our study consumed 0.12 g/g body weight (0.07) and 0.07 g/g body weight (0.04) for each 24 h restraint, respectively [1]. Whether this difference in food consumption is attributable to the different mouse strains or to the consistency of the presented diet needs to be investigated. We further assessed the lean and fat mass changes during restraint in both metabolic cages to examine body weight change in more detail. For both sexes, lean mass loss was more pronounced than fat mass loss suggesting that cold-induced shivering thermogenesis, referring to skeletal muscle tissue, predominates during 24 h metabolic cage restraint [6,16]. As for the reduction in body weight, lean mass was significantly decreased after TMC housing as opposed to the IMC.

To analyze cold stress more precisely, thermal images of mice in tested cage systems were taken at the end of each restraint. Our data clearly showed an increased cage temperature in the IMC compared to the TMC, which reflects the positive effect of the reduced IMC volume on the cage temperature regulation. Speakman and Keijer (2012) suggest that the room temperature for single-housed mice should be in the range from 23 °C to 25 °C [8].

The IMC temperature after the first and second restraint for female and male mice lies within this range (female—1st restraint: 23.7 °C; 2nd restraint: 23.5 °C; male—1st restraint: 23.3 °C; 2nd restraint: 23.5 °C). Whether the proposed room temperature range is sufficient in terms of cold stress reduction during 24 h metabolic cage housing remains a question to be answered. The room temperature in the experimental room of our study was kept constant at 23 °C (1). Conducting experiments at different room temperature intervals would therefore represent the next step [20]. We additionally assessed the body surface temperature besides cage temperature. Measurement of the body surface temperature was chosen over e.g., rectal temperature since it represents a non-invasive method that can be realized with use of a thermal imaging camera. Body surface temperature was significantly elevated for females and males after the second restraint in the IMC compared to the TMC. The body surface temperature of laboratory mice assessed via infrared thermography can serve as a surrogate for body core temperature and its potential to detect hypothermic conditions in mice on a regular basis could be exploited [14,36]. Thermal images in this study were taken between 08:00 a.m. and 10:00 a.m., which represents the inactive phase of mice. Mice possess higher heat insulation due to piloerection at inactive time intervals, while a greater surface area of a mouse is exposed to the surrounding environment during physical activity [8]. The concept of the thermoneutral point was introduced in addition to the thermoneutral zone describing a specific ambient temperature below which energy expenditure increases and above which body temperature increases [35]. The same authors showed a diurnal change in the thermoneutral point for C57BL/6J mice by 4 °C between the light and dark phase, also entailing changes in body temperature [35]. Because of the reported fluctuations concerning the body temperature of mice, depending mainly on mobile and immobile phases, assessing the animals’ body surface temperature on a frequent basis might have provided more insights [35]. We further investigated if the process of cold-induced non-shivering thermogenesis is initiated in mice restrained in metabolic cages by quantifying the mRNA expression of *Ucp1* in BAT [6,16]. For both sexes, no statistically significant differences in BAT *Ucp1* mRNA expression among the tested cage types were detected. Heat production in the context of non-shivering thermogenesis, which is realized by *Ucp1* located in BAT, can be exclusively used for maintaining the body temperature [16]. At the end of the second restraint, the body surface temperature, serving as a surrogate for core body temperature, in the IMC was significantly elevated compared to TMC. This applied to both sexes. *Ucp1* deficient mice on a C57BL/6J background were sensitive to cold exposure (5 °C) resulting in a reduction in body temperature, emphasizing that the ambient temperature represents a key variable directly affecting physiological pathways such as cold-induced thermogenesis in BAT [37,38]. In the case of chronic cold exposure, an adaptation process of thermogenesis was identified as non-shivering thermogenesis in BAT replacing shivering thermogenesis in skeletal muscle [38]. Thus, the absence of effects at the *Ucp1* mRNA expression level in the study performed here can be deduced with respect to the short 24 h period of metabolic cage restraint as well as the warm ambient room temperature of 23 °C (1). The fur score as an objective scoring system was applied to investigate the fur condition of mice, i.e., animal welfare, at the end of metabolic cage restraints. The fur score was significantly higher for female and male mice in the TMC and IMC compared to control cages after both restraints. In the context of another study, it could also be shown that the fur score is elevated after restraint in the TMC compared to controls [1]. The fur score of the TMC experimental group amounted to approximately 3 and a fur score of 1 referred to the control group, which provides good comparability to our data by assigning a clearly higher fur score for mice in the TMC [1]. First, it was suggested that an increased fur score could not solely represent a sign of decreased grooming, indicating impaired welfare, but more importantly, it is a way that mice are coping with the cold by raising up their fur hair for better heat insulation [1]. Second, mice might stop grooming in order to effectively save energy for the maintenance of body temperature.

Male C57BL/6J mice excrete about 73% of corticosterone metabolites via feces whereas females excrete about 53% [27]. We therefore suggest that excreted feces during a stressful intervention represent a suitable matrix. Female and male mice housed in the TMC excreted significantly higher fecal corticosterone metabolite (CM) concentrations during both 24 h restraints compared to animals housed in control cages. Here, the data from the females clearly showed that the TMC represented the greater stressor compared to the IMC since fecal CM concentrations for the IMC and control cages were in the same order of magnitude. A ten times higher fecal CM output was published for TMC housing compared to the control group while our data indicated an approximate two-fold increase with respect to both sexes and restraints [1]. In females, the IMC contributes to a noticeable reduction in stress because fecal CM levels after both restraints were comparable between the IMC and controls. Concerning male mice, fecal CM levels after the first restraint in the IMC were significantly higher compared with controls, which could be related to a sex-specific stress response.

In the next step of our analyses, the measured fecal CM concentrations were related to the assigned fur scores of each mouse. A fur score of 1 does not describe an impairment of welfare, which is why a fur score of 1 was set as the threshold value. A low fur score (=1) matched low fecal CM concentrations while a high fur score (>1) matched higher fecal CM concentrations, indicating that the fur score represents a reliable and easily assessable marker for the animal’s condition in terms of (cold) stress.

Our study demonstrated that the stress induced by experimental housing conditions is particularly important to consider in the course of data analysis. Therefore, the refinement of experimental procedures as well as the refinement of housing conditions are equally important. Research of this kind supports the protection of animal welfare during experiments, which concomitantly ensures the validity of the gathered data. The IMC represents a basis for further modification of the metabolic cage construction, but also an example of how experimental housing conditions can be refined in general. In order to complete the data, the food and water consumption should not only be recorded in the IMC and TMC, but also in the control cages. Defecation was assessed in all three tested cage types while urination was solely assessed in both metabolic cages. Hydrophobic sand could be used for urine collection in the control cage and to investigate whether this is a useful alternative to metabolic cage restraint for mice, as it was already described for rats [39,40]. To collect more data as part of metabolic phenotyping, the mean arterial pressure and heart rate of mice restrained in different metabolic cage types could be assessed [11]. Since implantation of a carotid artery catheter is required for this measurement, including anesthesia and surgery before the start of the experiment, we left this quite interesting measurement aside in the sense of the refinement guiding principle. There is still a need for improvement regarding the cage floor of the IMC. The decreased mesh size has evidently contributed to a pain reduction in the paws, but not all fecal pellets fall through the metal grid. Importantly, the amount of stuck fecal pellets accounts for a small percentage of the total per 24 h IMC restraint. Since we could not detect any regulation of *Ucp1* at the mRNA level after 24 h single-housing of mice among tested cage types, an investigation at the protein level would represent the next step. As part of our study, we examined the effect of cold stress during metabolic cage restraint on physiological processes including stress hormone production, non-shivering thermogenesis, and energy balance. Another key objective is to understand whether a behavioral response is triggered, which can be examined by means of behavioral tests, and whether this can also be quantified at the molecular level in the course of neurotransmitter analyses in the brain. We expect future research to focus not only on alternatives to animal testing but to also actively contribute to the refinement of housing conditions of laboratory animals, such as metabolic cage restraint, within the framework of indispensable animal testing.

## 5. Conclusions

Our data clearly demonstrate the considerable stress potential emanating from metabolic cage exposure on single-housed mice. Introducing the innovative metabolic cage (IMC) showed beneficial effects on mouse welfare by mainly alleviating cold stress during a 24 h restraint and by also taking the sex of mice into account.

Based on the improved IMC construction, it can be hypothesized that mice have to spend less energy in the IMC in contrast to the commercially available Tecniplast metabolic cage (TMC) to cover their body temperature requirements. As the IMC contributes to stress reduction, the variability of the collected data therein decreases concomitantly, also implying a significant improvement in data validity. Thus, when high animal welfare standards are applied with the use of the IMC, in turn, a study with greater certainty can be performed.

## Figures and Tables

**Figure 1 animals-13-02866-f001:**
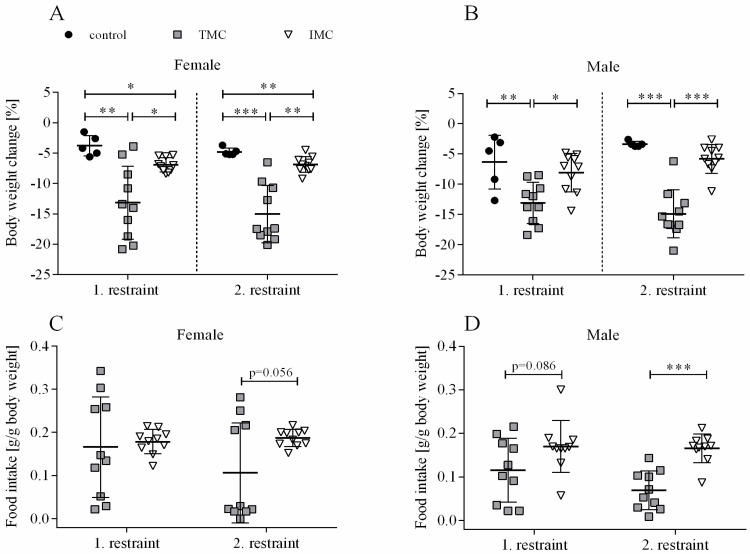
Body weight change and food intake of female (**A**,**C**) and male (**B**,**D**) mice. Female and male C57BL/6J mice were single-housed in one of the three cage types (control cage, Tecniplast metabolic cage, or innovative metabolic cage). The restraint was limited to 24 h and solely repeated once, totaling two restraints. Control, control cage (n = 5); TMC, Tecniplast metabolic cage (n = 10); IMC, innovative metabolic cage (n = 10). All data are presented as means (standard deviation). Differences between TMC and IMC, TMC and control, and IMC and control were calculated by one-way ANOVA (**A**,**B**). Differences between TMC and IMC were calculated by independent sample t-test (**C**,**D**). * *p* < 0.05, ** *p* < 0.01, *** *p* < 0.001.

**Figure 3 animals-13-02866-f003:**
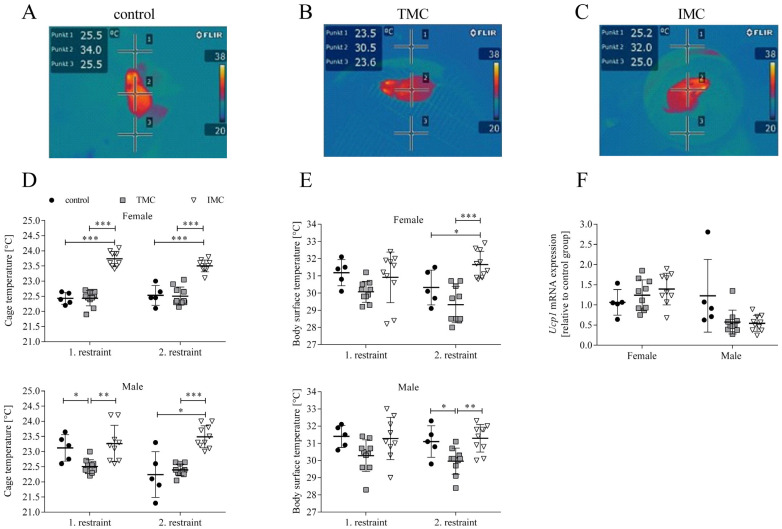
The innovative metabolic cage positively contributes to the maintenance of higher cage and body surface temperatures of mice. Thermal images of mice were taken during restraint in the control cage (**A**), Tecniplast metabolic cage (**B**), and innovative metabolic cage (**C**). Points 1 and 3 were used for cage temperature assessment while point 2 indicated the body surface temperature of the mouse. Cage (**D**) and body surface (**E**) temperatures for female and male mice at the end of each 24 h single-housing period in control cages (n = 5) and both metabolic cage types (n = 9/10). Relative mRNA levels of uncoupling protein 1 in interscapular brown adipose tissue of mice after the second restraint in three different cage types (**F**). mRNA levels of uncoupling protein 1 of mice in the Tecniplast and innovative metabolic cage (n = 9/10) were referenced to control mice (n = 5). Control, control cage; TMC, Tecniplast metabolic cage; IMC, innovative metabolic cage; *Ucp1*, Uncoupling protein 1. All data are presented as means (standard deviation). Differences between control and TMC, control and IMC, and TMC and IMC were calculated by one-way ANOVA (**D**–**F**). * *p* < 0.05, ** *p* < 0.01, *** *p* < 0.001.

**Figure 4 animals-13-02866-f004:**
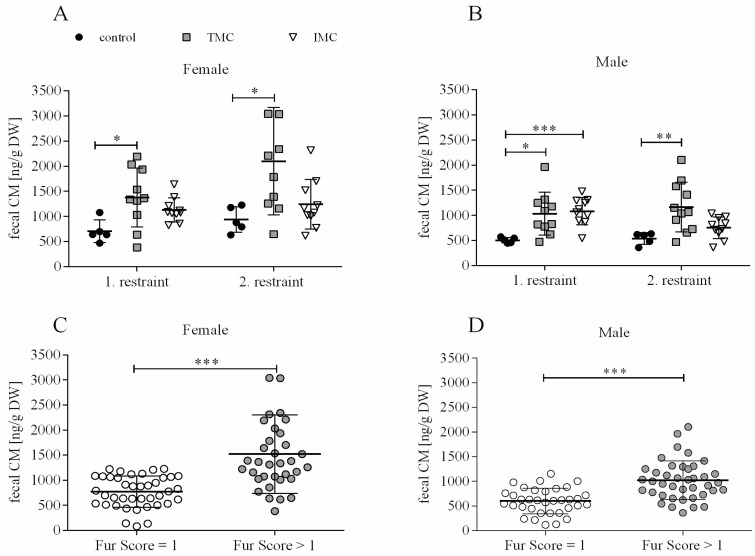
Fecal corticosterone metabolite excretion of female (**A**) and male (**B**) mice and fecal corticosterone metabolite concentrations in female (**C**) and male (**D**) mice matched to fur scores. Female and male C57BL/6J mice were single-housed in one of the three cage types (control cage, Tecniplast metabolic cage, and innovative metabolic cage). The restraint was limited to 24 h and solely repeated once, totaling two restraints. Control, control cage (n = 5); TMC, Tecniplast metabolic cage (n = 10); IMC, innovative metabolic cage (n = 10); CM, corticosterone metabolites; DW, dry weight. Differences between control and TMC, control and IMC, and TMC and IMC were calculated by one-way ANOVA (**A**,**B**). Fecal CM concentrations in female (n = 75) and male (n = 75) mice matched to fur scores for each mouse including all points in time (baseline, 1st, and 2nd restraint) as well as cage types (control cage, Tecniplast, and innovative metabolic cage). Fur scores were subdivided into two groups: fur score = 1 and fur score > 1. Differences between fur score categories (**C**,**D**) were calculated by an independent sample t-test or Mann–Whitney U test. All data are presented as means (standard deviation). * *p* < 0.05, ** *p* < 0.01, *** *p* < 0.001.

**Table 1 animals-13-02866-t001:** Humane endpoints: termination criteria for the conducted experiment.

Criteria for Animal Welfare Assessment	Score
Reduced grooming	1
Body weight loss < 5%	
Feces: slight changes in shape (pasty, still shaped), consistency (soft), smell, color	
Significantly reduced grooming	2
Gummy eyes	
Blocked nose	
Fecal contamination	
Body weight loss 5–15%	
No grooming	3
Half-closed, pale, gummy eyes	
Curved back	
Crouched posture	
Positive skin fold test	4
Feces: absence of defecation, major change in shape (unformed)/ consistency (liquid)/striking smell or color	
Body weight loss 20%	5
Disorientation	

**Table 2 animals-13-02866-t002:** Fur scores of female and male mice after metabolic cage restraint compared to mice housed in control cages.

Cage Type	No. of Restraint	Mean Fur Score (Standard Deviation)		Statistics		
		Female n = 25	Male n = 25		Female n = 25	Male n = 25
control	baseline	1.00 (0.00)	1.00 (0.00)	control vs. TMC	*p* = 1.000	*p* = 1.000
TMC		1.00 (0.00)	1.00 (0.00)	control vs. IMC	*p* = 1.000	*p* = 1.000
IMC		1.00 (0.00)	1.00 (0.00)	TMC vs. IMC	*p* = 1.000	*p* = 1.000
control	first	1.00 (0.00)	1.00 (0.00)	control vs. TMC	*******	*******
TMC		2.00 (0.00)	2.75 (0.26)	control vs. IMC	*******	*******
IMC		2.00 (0.00)	2.00 (0.00)	TMC vs. IMC	*p* = 1.000	*******
control	second	1.00 (0.00)	1.00 (0.00)	control vs. TMC	*******	*******
TMC		2.00 (0.00)	2.50 (0.53)	control vs. IMC	*******	*******
IMC		2.00 (0.00)	2.00 (0.00)	TMC vs. IMC	*p* = 1.000	*p* = 0.063

Female and male C57BL/6J mice were single-housed in one of the three cage types (control cage, Tecniplast metabolic cage, or innovative metabolic cage). The restraint was limited to 24 h and solely repeated once, totaling two restraints. The fur scores of female and male mice at baseline and after the first and second restraint in control cages (n = 5) and both metabolic cage types (n = 10). Control, control cage; TMC, Tecniplast metabolic cage; IMC, innovative metabolic cage. All data are presented as means (standard deviation). Differences between cage types for each point in time were calculated by the Mann–Whitney U test. *** *p* ≤ 0.001.

## Data Availability

The data that support the findings of this study are available from the corresponding author upon reasonable request.

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
