# Peer review of "Reduction in Cold Stress in an Innovative Metabolic Cage Housing System Increases Animal Welfare in Laboratory Mice"

_animals, 2023, doi:10.3390/ani13182866_

Round 1

Reviewer 1 Report

I would like to congratulate excellent work and thank you for a fascinating reading. In my opinion Your text can potentially be considered a template for further Authors aiming to master scientific reports on laboratory animals maintenance and welfare. Best wishes!

I like this manuscript a lot. It can be seen that the Authors put an effort in the experimental phase and in manuscript preparation.

In my opinion the main question posed by the Authors is: can we improve the conditions of laboratory animals by putting attention to important technical details of maintenance equipment? And this manuscripts is of a great help in this matter. Yes, I believe we can perceive it as highly relevant and extremely interesting for readers.

I believe it is a solid scientific work and would expect an increased interest form both readers and future Authors (citations). It definitely states a firm point for continued comparisons and research.

It was a great pleasure to read the text. For me it is an example of an excellent scientific writing.

I think that all the conclusions proposed by the Authors are valid and greatly address aim of the study.

Reviewer 2 Report

Refinement of any procedure performed on research animals is important to improve their welfare. Current topics include assessing the effect that routine practices have on animals or establishing the impact that common procedures (such as the use of metabolic cages) have on rodents. The present article presents an innovative approach to address this issue, comprising interesting physiological, behavioral, endocrine, and even genetic findings. However, some parts of the manuscript need improvements. As a general comment. The beginning of the manuscript is a little confusing because several elements that are part of the experimental design are missing. For example, in the simple summary and abstract, there is no mention of what parameters the authors used to evaluate stress (e.g., body weight, fecal metabolites, food intake, etc.). Then, only at the end of the introduction, there is mention of more parameters such as fur condition, thermal imaging, or mRNA expression in BAT. These elements need to be mentioned in the Introduction. Explain why thermal imaging can be used to assess cold stress, or why mRNA is used to determine BAT activation in hypothermic animals.

Simple summary. After stating the aim of the study, I recommend briefly adding what parameters were evaluated to determine that the innovative metabolic cage diminished stress in mice (e.g., concentration of fecal corticosterone metabolites, body weight, etc.).

Response:

Lines 26-30. Please, clearly state that three housing cages/experimental groups were compared. My observation is because lines 26-28 state that IMC was compared with TMC but lines 29-30 mention that animals were housed in control, IMC, and TMC cages, and this is also mentioned in the Methods section (three experimental groups). Also, similarly to my comment for the Simple summary, include which parameters were evaluated to determine that mice were more/less stressed (e.g., body weight, fecal metabolites).

Response:

Lines 30-32. When mentioning that body weight decreased less or that higher FCM was found, it would be helpful to include the actual values in grams or mg/dl to understand this difference.

Response:

Line 54. I recommend starting a new paragraph from “Cold stress therefore describes…”. The same applied to line 62 “The energy expenditure of mice may be elated…” and line 79 “Thermoneutrality depends on many factors including…”.

Response:

Line 65. Add a reference.

Response:

Lines 79-80. Size is also important and might be a reason that could explain why mice are more sensitive than rats to cold-induced stress. Mice being small animals have a larger surface area in relation to their volume. Therefore, they lose more heat and need to use greater metabolic energy to produce heat. These references might help the authors: Lezama-García et al., 2020 https://doi.org/10.3390/vetsci9050246 and Payne et al., 2018 https://doi.org/10.1002/ajpa.23432.

Response:

Line 86-88. Consider moving these lines (the reason why authors selected this mice strain) to “Animals and housing condition”. Also, consider removing lines 88-89 if this information is already included in the Methods section.

Response:

Lines 94-96. Following my observation at the end of the general comment, up until this point, there is no mention of thermal imaging or mRNA expression levels. To understand why the authors evaluated all these parameters, the reader must see in the Introduction why thermal imaging can be used as a method to assess thermoregulatory mechanisms of animals (see Mota-Rojas et al., 2021 https://doi.org/10.3390/ani11061733 ) or why the expression of UPC1 and mRNA in BAT has been used or is relevant for cold-induced stress or hypothermia (see Bienboire-Frosini et al., 2023  https://doi.org/10.3390/ani13132173).

Response:

Line 98. This observation applies to all subheadings in the manuscript. Please, revise the journal’s template. Each subsection needs to be numbered. For example: 2.1. Ethical statement, 2.2. Animals and housing conditions, 2.3. XXXX.

Response:

 Line 109. Include the sample size. Also, mention what were the inclusion/exclusion criteria for these animals.

Response:

Lines 136-137. Did the authors compare two or three cages? Because in the Abstract it is stated that a control cage, IMC, and TMC were evaluated. In lines 163-164 three experimental groups are described: control, TMC group, and IMC group.

Response:

Lines 208-213. More detail is needed for the Body composition evaluation. This was performed inside each evaluated cage? At what period?

Response:

Line 216. Consider replacing the word “photos” with “radiometric images” or “thermal imaging”.

Response:

Lines 221-224. More detail is needed (or perhaps a thermal image per se) to understand the delimitation of the thermal imagining. “The back of the mouse” is not very specific and the reader cannot understand if the authors evaluated, for example, solely the interscapular region (located in the back), or if the temperature considered from the neck to thoracic vertebrae region.

Response:

Line 269. Please, delete “(2003)”.

Response:

Lines 311-313. I consider that these lines fit better in the Methods section than in the Discussion. Consider moving/deleting them to the Methods. This comment also applied to line 324.

Response:

Line 507. Please, adapt the Table format according to the journal’s template.

Response:

Lines 568-574. The aim mentioned in these lines is more complete and easier to understand that the aim mentioned in the Simple summary, Abstract, or Introduction. Consider maintaining this description in other places as well. Also, before the discussion and association between the findings and the published literature, a brief overview of thermoregulation and physiological responses of animals to cold stress would help to understand why body weight, surface temperature, food intake, and other parameters showed differences according to the housing cage (Mota-Rojas et al., 2021 https://doi.org/10.3390/ani11061733).

Response:

Line 594: Please, consider adding the actual values found for body weight, lean mass, temperature, and fecal corticosterone metabolite concentration.

Response:

Line 613. Amend the in-text citation style in this line and throughout the discussion section.

Response:

Lines 647-651. Here, the authors say that thermal imaging can serve as a “standardized surrogate for body core temperature”, but this is not always true. It highly depends on the species and the analyzed region. For example, some studies have determined that analyzing the ocular surface temperature is correlated with body temperature changes, while other authors mention body surface temperature. Additionally, depending on the exact point where the surface temperature was evaluated in the present study (which needs to be included in the Methods), the increased surface temperature found in IMC could be considered as a higher stress response due to a greater activation of BAT to compensate for cold stress? Knowing that UPC1 at BAT is activated by norepinephrine and that norepinephrine is released when animals are stressed. Consider discussing this information.

Response:

Line 705. Before conclusions, please, include the limitations of the study and further recommendations according to the findings of the present study.

Response:

 Decision: Accepted with major changes.

Reviewer 3 Report

This manuscript describes a modified metabolic cage for mice, aimed at reducing stress during metabolic cage housing. Some measured parameters (phenotypic data, cold stress by thermal imaging and fecal corticosterone metabolites) indicate some advantages of the new metabolic cage over a commercially available metabolic cage.

The article is of potential interest as it may be of help in the design of innovative housing systems for mice (and rodents in general). Furthermore, it is comprehensive and presented in a well-structured way. However, the authors should provide more essential information.

COMMENTS AND QUESTIONS

Materials & Methods

Animals and housing conditions

It is not clear to me what type of cages were used to house the animals immediately prior to the study. The authors mention that the mice were housed in conventional type III cages, but immediately afterwards they say that males are housed alone, in smaller cages (type II), because of their aggressive behavior. Does this mean that only females are housed in type III cages? Please clarify this point for a better understanding.

The description of the environmental conditions is very detailed and precise, but I miss information related to the system of switching on and off the lights, to know if it is done in an abrupt or progressive way (dusk – dawn system). I think this could further improve the description of the environmental conditions.

Study design

The authors say that, after evaluating the Fur Score, the animals are returned to their home cages with familiar group constellations. Again, it is not clear to me if this refers to the male mice as well.

There seems to be a missing verb in the sentence “After the first restraint into different cage systems, a 6 d resting period followed by a second restraint according the same experimental workflow.” (lines 178-180).

Humane endpoints: termination criteria

The authors mention that a score of 2 is a termination criterion and leads to euthanasia, and that none of the animals used in the study met the predefined termination criteria. Table 1 scores a body weight loss of 5-15% as 2. However, looking at the individual values in Figure 1A and B (first restraint), almost all animals have undergone a body weight change of 5% or more (and more than 20% in the case of 2 females). It appears that none of these animals have been euthanized, as the same n=10 is present in the 2nd restraint graph. Please clarify this point.

Food intake

Please justify why the food consumption of the control animals was not monitored. I think this would have been interesting for the study.

Expression of Uncoupling protein 1 in brown adipose tissue

Please use italics for E. coli.

Results

If the authors have monitored the amount of feces and urine collected in both types of metabolic cages, it would be interesting to show the results, to see if there is any difference between both designs. The search for a housing system that reduces stress for the animals should not overlook the fact that the main function of metabolic cages is the careful and accurate collection of feces and urine. In particular, I would have liked the authors to show that changing the type of grid on which the animals lie does not interfere with the number of fecal pellets collected.

3.1. Changes in body weight, body composition, and food intake are dependent on the metabolic cage type

If we look at the data shown in figure 1C, the females eat practically the same in IMC, both in the first and in the second restraint. In reality, what seems to happen (and what explains the statistically significant difference observed) is that females in TMC eat less in the second restraint. Therefore, it seems better to express this observation as "females showed a tendency to consume less food during the second restraint in TMC than in IMC". As mentioned above, it would have been very useful to have the values of the control animals to make a more accurate comparison.

Lines 398-400: I do not fully agree with this sentence, similarly to what was explained just before. In my opinion, what is apparent is a decrease in food intake in TMC.

3.3. Excretion of corticosterone metabolites in the feces is decreased by restraining mice in the 516 innovative metabolic cage

Nowhere in the text has it been previously indicated what the acronym DW, which is mentioned extensively in this section, corresponds to. Please indicate.

Discussion

In my opinion, the authors should include in the discussion the potential influence of housing males in solitary confinement prior to the study. Females, in contrast to males, may suffer higher stress since solitary housing is a novelty. Males, on the other hand, may have lower stress values because they are usually housed in solitary confinement.

At no point do the authors explain the reason for a second restraint. Under experimental conditions, mice are usually housed only once in metabolic cages, so the second restraint period would not make much sense. Even if this double restraint would be useful for the authors because they could usually perform it in their facilities, the differences found between one restraint and the other in the different parameters studied are not sufficiently discussed.

Last paragraph: Although the authors argue that the nearly equal fecal CM concentrations for the controls and IMC in females indicate that TMC represents a major stressor, they do not explain the significance of the nearly identical levels of corticosterone in males in TMC and BMI.

Conclusions

It might be appropriate to mention here that the potential beneficial effects of a new metabolic cage design should take into account the sex of the animals, in view of the variations between sexes observed in some parameters of the study.

Tables and Figures

Table 2

I would like to suggest (and the same applies to the following figure captions) that the first two sentences be deleted, as their content is already sufficiently explained in the article and is repetitive. Therefore, the table caption would start with "Fur Scores of female and male mice...".

Figure 1

Same as above. Therefore, the figure caption would start with "Body weight change of female (A) and male (B) mice...".

Figure 2

Same as above. Therefore, the figure caption would start with "Lean mass change of female (A) and male (B) mice...".

Figure 4

Same as above. Therefore, the figure caption would start with " Fecal corticosterone metabolite concentrations of female (A) and male (B) mice...".

Please indicate the meaning of DW in the figure caption for better understanding of the figure by itself.

Round 2

Reviewer 2 Report

The authors diligently addressed each of my observations step by step.

The manuscript is now more seamlessly integrated and better comprehensible. I have no further comments to add.

The article should be published

Reviewer 3 Report

This second version of the manuscript is a significant improvement over the previous one. Studies like the present one can be of great help to improve housing systems as complex and necessary for research as metabolic cages (which at the same time are particularly problematic from the point of view of animal welfare).

Thank you very much for the excellent job.